# Dynamics of major plant nutrients and enzymatic activities in soil influenced by application of biochar and organic waste

M. L. Dotaniya[1]*, M. D. Meena[1]*, R. L. Choudhary[1], M. K. Meena[1], V. D. Meena[1], Harvir Singh[1], Brij Lal Lakaria[2], R. S. Jat[1], P. K. Rai[1], Kuldeep Kumar[3], R. K. Doutaniya[4], Harpreet Singh[5]

**1** ICAR-Directorate of Rapeseed- Mustard Research, Bharatpur, India, **2** ICAR- Indian Institute of Soil and Water Conservation, Research Centre, Chandigarh, India, **3** ICAR- Indian Institute of Soil and Water Conservation, Research Centre, Kota, India, **4** Department of Agronomy, SKN College of Agriculture, Jobner, India, **5** Regional Research Station, PAU, Gurdaspur, India

* mohan30682@gmail.com (MLD); murliiari@gmail.com (MDM)

**Data Availability Statement:** All relevant data are within the manuscript file.

**Funding:** The author(s) received no specific funding for this work.

## Abstract

The concentration of salt ions influences the availability and plant nutrients dynamics in the soil. Proper management of these ions can enhance food grain production, helping to feed the growing population. In this experiment, nine fertility combinations were followed to enhance the soil organic carbon and reduce the salt toxicity and monitor the plant nutrient availability. An incubation experiment was conducted for the period of one year with different organic soil amendments in combinations including biochar (BC), pressmud (PM), and farm yard manure (FYM) as follow: $T_1$-control, $T_2$-RDF, $T_3$-FYM (10 t/ha), $T_4$-PM (10 t/ha), $T_5$-BC (10 t/ha), $T_6$-FYM (5 t/ha) + PM (5 t/ha), $T_7$-FYM (5 t/ha) + BC (5 t/ha), $T_8$-PM (5 t/ha) + BC (5 t/ha), $T_9$-FYM (5 t/ha) + BC (2.5 t/ha) + PM (2.5 t/ha). Results showed that addition of organic substance (10 t/ha) significantly ($p < 0.05$) affected soil pH and electric conductivity. Plant nutrient availability (N, K, and S) was also influenced by application of organic substance (10 t/ha). Organic C and available N were recorded the highest in the treatment $T_7$ (FYM—5 t/ha + BC -5 t/ha); whereas, the highest available K and S were observed in treatment $T_5$ (BC-10 t/ha). The microbial soil fertility indicators (alkaline phosphatases, arylsulphatase, dehydrogenase activity and microbial biomass carbon) were measured the highest in FYM (5 t/ha) + BC (5 t/ha) applied treatment. In conclusion, application of organic substance 10 t/ha (biochar alone or with FYM) improved the plant nutrient availability and soil microbial activities in saline soil. It could be a suitable option for enhancing the soil fertility in saline soils.

## 1. Introduction

The population of developing countries is increasing with time and require huge amount of food stuffs. By the year of 2050, Indian population (1.74 billion) will require approximately 350 million tonnes (mt) of food grains [1]. Indian soils are having low to medium range of soil

**Competing interests:** The authors have declared that no competing interests exist.

organic carbon (SOC) and greatly influence the nutrient use efficiency and enhance the benefit: cost ratio in most of the cropping systems [2]. The addition of organic substances is the key to enhancing SOC. This can be achieved by applying crop residues, vermicompost, green manure, municipal solid waste and FYM [3,4]. The availability of FYM is going down and farmers are mostly using straight or complex fertilizers for crop production [5]. It leads to poor crop yield and declining trend of soil fertility parameters over a period [6,7]. However, farmers are also using the pressmud (PM) (a byproduct of sugar industry) in sugarcane growing areas. It is having significant amount of carbon (C) and plant nutrients. In India, annually 11.4 mt of PM is available for its potential use [8]. It improves the SOC status and plant nutrient dynamics in most of the soils [9]. To improve soil health parameters and crop yield in soils with poor health, low soil SOC, poor fertility, excessive salt, metal toxicity, and ravine soils, it is necessary to apply organic matter [10,11]. The organic substances improve the soil microbial population and diversity, leads the better mineralization of soil organic matter (SOM), release different type of organic acids enhance the solubilization of many nutrients in soil [12,13].

Biochar is becoming increasingly popular for enhancing soil fertility and crop production. As a carbon source produced through slow pyrolysis, its physico-chemical properties are significantly influenced by the nature of the organic materials used, the temperature of the process, and the oxygen supply during production. Lehmann and Joseph [14] pointed out that application of biochar improved the soil fertility parameters, moisture content, plant nutrient availability, soil microbial biomass count and diversity. Biochar operates on the principle of adsorption kinetics, absorbing plant nutrients from the soil and making them easily available to plant roots. These adsorption processes are also crucial for the remediation of pollutants from soil and water systems [15]. Apart from this, increasing the climatic change parameters mainly an atmospheric temperature leads the efficiency of the natural resources and minimizes the crop yield potential over a period. Incorporation of biochar enhanced the C sequestration potential of soil to combat the climate change effect [16]. Biochar is a potential source of plant nutrients, energy, solution to climate change, and popular for carbon negative energy production [17]. Saline soils have limitations due to excessive salt, which reduces water and nutrient uptake in most crops. Applying biochar at a rate of 5–10 g/kg has been shown to improve soil nutritional status, cation exchange capacity, and nutrient storage capacity, while also reducing soil acidity in Gleysols [18]. It has also improved the macro and micronutrients status in soils and reduced greenhouse gases emissions from soils [19].

The assessment of organic substances amended soil condition is effectively gauged through enzyme activity, making it a prime metric. Mandal et al. [20] reported that FYM and poultry manure along with the chemical fertilizers improved the soil organic C, plant nutrient status in soils. While microorganisms predominantly contribute to the production of enzymes in soil, it's noteworthy that plant roots and soil-dwelling animals can also contribute to the release of these enzymatic compounds [21]. Soil enzymes can detect changes in soil management practices well before other indicators of soil quality. These enzymes, crucial for decomposing organic materials and facilitating nutrient cycling, show significant responses to changes in soil conditions. Implementing soil management practices and introducing organic amendments enhance soil organic matter, which in turn elevates enzyme activity. Specifically, the carbon-to-sulphur ratio in the soil is modulated by the introduction of organic sources, which boosts the activity of arylsulphatase. Arylsulphatase serves as an indicator of sulphur (S) levels in the soil, showing a direct correlation with labile S in the soil solution, as evidenced by research [22].

Soil enzymes expedite the decomposition of plant residues, releasing nutrients that become readily available to plants. Consequently, the cumulative microbial activity over an extended period, along with the viability of the microbial population at the time of sampling, is referred

to as enzyme activity. Dehydrogenase (DHA), a crucial intracellular enzyme of microbes, plays a vital role in governing the transfer of electrons from organic substances to inorganic acceptors. Consequently, it has a significant impact on the breakdown of organic matter within soil, as highlighted by Wolińńska and Stępniewska [23]. The quantification of soil microbial biomass carbon (MBC), proposed as an indicator of soil stress and disturbance by Hernández et al. [24], holds significant importance in soil ecology research. Microbial activity, through the recycling of nutrients and energy, plays a vital role in indirectly contributing to ecosystem function. However, unfavorable anthropogenic effects can induce stress conditions, leading to abnormal alterations in microbial diversity and the biologically active components of soil organic matter. These changes encompass variations in microbial biomass, enzymes, and various organic compounds, such as proteins or carbohydrates [25].

In this backdrop, a hypothesis was formulated to measure the plant nutrient content and soil enzymatic activities in saline soil mediated through organic substances addition (biochar, PM and FYM).

## 2. Materials and methods

### 2.1. Location

The experiment was carried out at the ICAR-Directorate of Rapeseed-Mustard Research, Bharatpur. It is located 77.300˚E longitude and 27.150˚N latitude with above mean see level of 178.37 meter. The summer months of Bharatpur are quite hot, with maximum temperature ranging 38˚C to 45˚C from March to June. During winters, temperature ranges from a maximum of 23˚C to a minimum of 7˚C in the month of January. The monsoon season (July to September) brings with it a drop in temperature to about 27˚C and a 70%–75% humidity level.

### 2.2. Soil sample collection and analysis

Soil samples were collected from saline patches of farmers field located in Bharatpur region of India. The collected soil was processed and kept for physico-chemical properties. With the help of standard soil analytical procedure, different parameters of soil samples were measured [26]. The soil is having pH (8.62), EC (2.32 dS/m), Walkley-Black carbon (0.33%), and a medium range of available N, P and K. Based on these data, different treatments were planned to improve soil organic carbon and plant available nutrients in soil.

### 2.3. Treatment details

In this incubation experiment, nine treatments comprised with biochar (BC), pressmud (PM) and farm yard manure (FYM) were applied *i.e.*, $T_1$-control, $T_2$-RDF, $T_3$-FYM (10 t/ha), $T_4$-PM (10 t/ha), $T_5$-BC (10 t/ha),$T_6$-FYM (5 t/ha) + PM (5 t/ha), $T_7$-FYM (5 t/ha) + BC (5 t/ha),$T_8$-PM (5 t/ha) + BC (5 t/ha), $T_9$-FYM (5 t/ha) + BC (2.5 t/ha) + PM (2.5 t/ha) (Table 1). The biochar, FYM and pressmud were analysed prior to application into the soil. In this experiment, pigeon pea biochar was applied as per the treatments. The process involved cutting the pigeon pea stem into pieces measuring 10–15 cm. After drying, the biomass underwent pyrolysis at 300˚C for 2 hours, followed by quenching and subsequent drying in an oven at 105˚C. The resulting biochar was then crushed using a 24-blade Rotar Mill (Model No. Pulversittee 14) and sieved to obtain uniform particles sized between 53 and 75 μm [27]. For the quality assurance/quality control (QA/QC), physico-chemical properties of biochar were determined in replicated samples, i.e. pH-8.1, EC (dS/m) 0.43, N 14.1 g/kg, P-0.52 g/kg, K-3.1 g/kg, C-662 g/kg; C/N ratio-58.6. Pressmud was also analysed before application and measured as pH (8.21), EC (3.18), potassium concentration (0.039%), sodium content (0.013%), OC (9.51 mg/kg),

**Table 1. Treatment combinations of different soil amendments applied during experiment.**

| Treatments | | Descriptions |
|---|---|---|
| T₁ | Control | Without fertilizer and organic manures |
| T₂ | RDF | Recommended dose of fertilizer for mustard (80:40:40:: N:$P_2O_5$: $K_2O$) |
| T₃ | FYM (10 t/ha) | Well mature FYM (10 t/ha) |
| T₄ | PM (10 t/ha) | Pressmud (10 t/ha) |
| T₅ | BC (10 t/ha) | Biochar (10 t/ha) |
| T₆ | FYM (5 t/ha) + PM (5 t/ha) | Organic substances (10 t/ha) (1:1 FYM:PM) |
| T₇ | FYM (5 t/ha) + BC (5 t/ha) | Organic substances (10 t/ha) (1:1 FYM:biochar) |
| T₈ | PM (5 t/ha) + BC (5 t/ha) | Organic substances (10 t/ha) (1:1 PM:biochar) |
| T₉ | FYM (5 t/ha) + BC (2.5 t/ha) + PM (2.5 t/ha) | Organic substances (10 t/ha) (2:1:1:: FYM:biochar: PM) |

total N (0.021%), total P (0.013 mg/kg), and S (0.056%). However, FYM properties such as pH (7.22), EC (1.59 dS/m), potassium concentration (0.043%), OC (14.12 mg/kg), total N (0.55%), total P (0.028 mg/kg) and S (0.016%) were analysed. With the help of preliminary experiment, doses of different organic substances were calculated based on the C content. The soil fertility parameters analysed to observe the different fertility ratios (C:N and C:P) mediated the nutrient dynamics in soil. The treatments were applied by mixing organic wastes thoroughly with soil and water holding capacity was maintained regularly by adding the deficit moisture. After one year incubation period, soil samples were taken, processed as per the soil samples analysis procedure, and analysed for soil physico-chemical properties [26]. The microbial soil health parameters were also analysed in soil. The alkaline phosphatase was measured by Tabatabai and Bremner [27]; dehydrogenase activity (DHA) was analysed by the method described Casida et al. [28]; arylsulphatase activities by Tabatabai and Bremner [29]; MBC as per the procedure described by Jenkinson and Powlson [30]. In this process, the MBC was assessed by subtracting the evolved $CO_2$-C from unfumigated soil from that of fumigated soil over a ten-day incubation period, then dividing the result by a $K_c$ value of 0.45.

## 2.4. Statistical analysis

The experiment was conducted in completely randomized design (CRD) with three replications. In this experiment combination of three organic soil amendments were comprised into nine treatments. The collective soil data was statistically analysed for pH, EC, OC, available nutrients (N, P, K, S) and enzymatic activities. Data were analyzed as per the procedure mentioned in Gomez and Gomez [31]. The analysis of variance (ANOVA) was calculated for all the treatments. The least significant difference (LSD) was calculated at 5 percent level of significance (p = 0.05).

## 3. Results and discussion

### 3.1. Effect on soil physico-chemical properties

The destructive soil samples were collected after one year. Initial soil pH was 8.61; and application of different organic treatments significantly affected the soil pH (Table 2). Application of FYM, PM and BC (10 t/ha) reduced the soil pH 8.58, 8.60 and 8.59, respectively. The electrical conductivity of soil 2.31 dS/m was also reduced to 2.12 dS/m in FYM, 2.14 dS/m in FYM (5 t/ha) + BC (5 t/ha) and FYM (5 t/ha) + BC (2.5 t/ha) + PM (2.5 t/ha) treated soils. Soil organic C is very important for enhancing the most of the soil health parameters. In this experiment,

**Table 2. Effect of different treatments on soil physico-chemical properties (n = 3).**

| Treatments | | pH | EC | OC | Av N | Av K | Av S |
|---|---|---|---|---|---|---|---|
| | | | dS/m | % | kg/ha | | |
| T$_1$ | Control | 8.61 | 2.31 | 0.319 | 148.2 | 123.5 | 12.4 |
| T$_2$ | RDF | 8.62 | 2.29 | 0.321 | 153.4 | 136.4 | 13.7 |
| T$_3$ | FYM (10 t/ha) | 8.58 | 2.12 | 0.451 | 189.7 | 165.5 | 17.4 |
| T$_4$ | PM (10 t/ha) | 8.60 | 2.16 | 0.385 | 175.6 | 144.4 | 17.8 |
| T$_5$ | BC (10 t/ha) | 8.59 | 2.15 | 0.456 | 182.3 | 177.3 | 18.2 |
| T$_6$ | FYM (5 t/ha) + PM (5 t/ha) | 8.59 | 2.18 | 0.437 | 178.6 | 152.1 | 16.7 |
| T$_7$ | FYM (5 t/ha) + BC (5 t/ha) | 8.55 | 2.14 | 0.459 | 194.2 | 168.4 | 17.1 |
| T$_8$ | PM (5 t/ha) + BC (5 t/ha) | 8.57 | 2.17 | 0.423 | 183.4 | 143.2 | 17.3 |
| T$_9$ | FYM (5 t/ha) + BC (2.5 t/ha) + PM (2.5 t/ha) | 8.54 | 2.14 | 0.454 | 187.8 | 161.5 | 17.9 |
| LSD (p = 0.05) | | 0.05 | 0.13 | 0.021 | 33.5 | 18.4 | 2.4 |

addition of different sources of C improved the soil organic C level of soil. It was observed that FYM applied treatment had 41% higher organic C than the control treatment. The highest SOC (0.459%) was measured in 10 t/ha organic residue applied through combination of PM (5 t/ha) + BC (5 t/ha). Data showed that the addition of C through the organic amendments improved the concentration of available plant nutrients during the incubation period. It was observed that initial level of available N (148.2 kg/ha) increased to 189.7 kg/ha in 10 t/ha FYM applied soils, which is 28% higher. In similar way, addition of 10 t/ha organic substance through PM and BC (equal part) enhanced available N to 194.2 kg/ha. It is the highest improvement in available N among the treatments.

Available potassium increment was recorded up to 37.3% in T5 -BC (10 t/ha), which is indicated that the biochar properties affected the mineralization and adsorption of K$^+$ ions (Fig 1). However, other organic treatments were also improved the available K in soil $i.e.$, 165.5, 168.4 and 161.5 kg/ha in FYM (10 t/ha), FYM (5 t/ha) + BC (5 t/ha), FYM (5 t/ha) + BC (2.5 t/ha) + PM (2.5 t/ha), respectively. Here the integration of FYM, BC and PM did not surpass the effect of BC alone as well as combination of PM + BC (in equal amount). Results of available S showed that addition of organic treatment combinations also improved the soil fertility parameters. The significantly higher (p = 0.05) available S (18.2 kg/ha) was measured in 10 t/ha BC treatment; whereas, PM applied @10 t/ha, and FYM (5 t/ha) + BC (2.5 t/ha) + PM (2.5 t/ha) showed at par response during the experiment.

The addition of organic substances through the FYM, PM and BC are key source of C. Most of the crop residues are containing 42–46 percent C except legume crops. The addition of these organic sources provides the food stuff for soil biota. However, the dynamics of the C mineralization in soil is greatly influenced by the soil, substances and climatic factors. Biochar is a partial oxidized material, having lot of the C and micro-capillary pores having the higher CEC leads to better exchange of plant nutrient between soil and plant roots. It enhance the soil C level for longer period, promote constant but limited supply to microbes, enhance the mineralization of available sources of C in soil [32]. However, poultry manure also contains carbon and small fractions of sugar, which promote microbial populations in saline soil [33]. Incorporating organic matter through the use of PM enhances soil microbial populations and diversity [34], thereby improving the mineralization kinetics of SOM and influencing the dynamics of available nitrogen in the soil. Moreover, microbial decomposition could release essential nutrients for plant uptake when readily available organic carbon is sourced from PM [35]. Conversely, the combined application of FYM and biochar results in a higher percentage and mixed pools of carbon, leading to a direct accumulation of carbon in the soil. The application

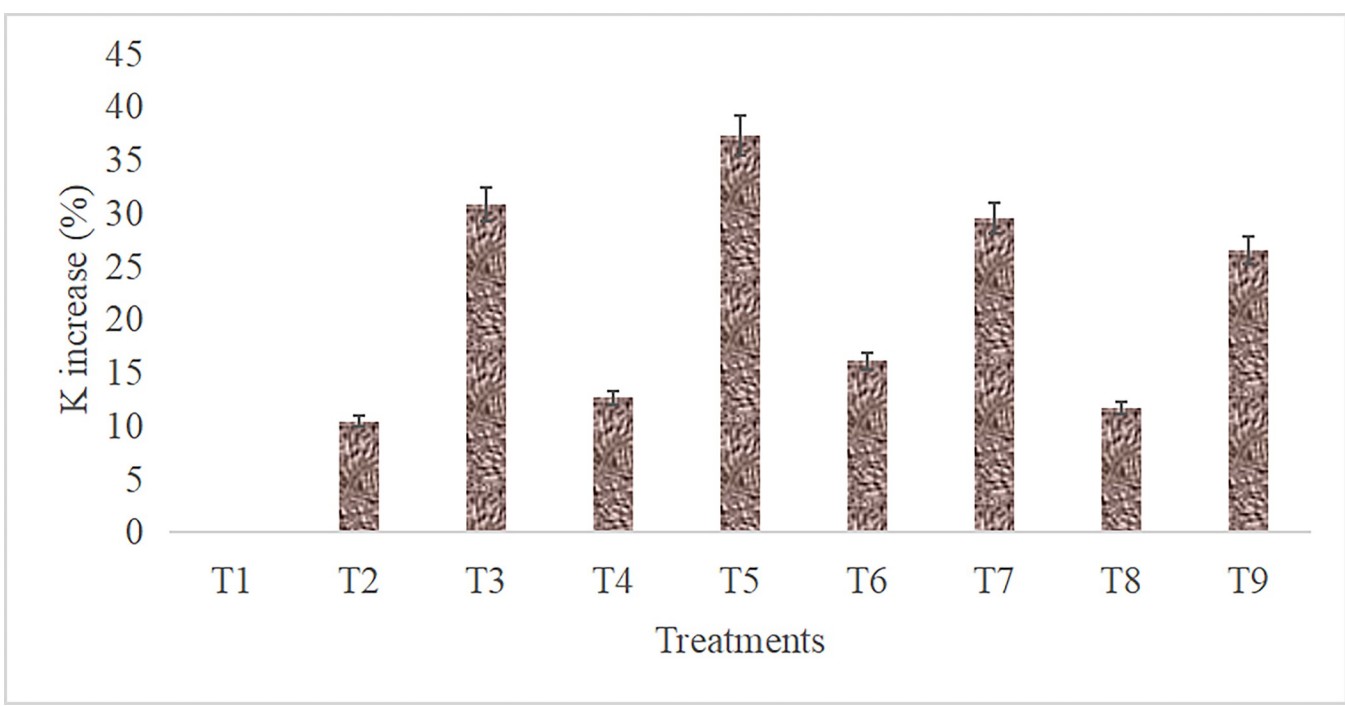

**Fig 1. Different treatment combinations affect the available K in soil (n = 3).**

of biochar stores carbon in the soil for longer duration, mainly due to its recalcitrant nature [36]. Additionally, applying biochar in saline soil adsorbs excessive amounts of sodium from the soil solution, providing more opportunities for the uptake of essential plant nutrients [37,38]. Singh et al. [39] conducted a study to monitor the effect of biochar (4 t/ha), and observed that biochar application improved the soil chemical (plant nutrients), physical (pH, EC) and biological process (microbial count) under saline ecosystems. Biochar is having higher number of pores, which leads the adsorption of plant nutrients and their release kinetics during plant growth [40]. Mineralization of organic substances releases various types of organic acids, which act as bio-absorbers in saline soil [41]. This process reduces salinity and enhances the availability of plant nutrients in the soil solution [42]. Additionally, the addition of organic carbon through biochar has been shown to improve the nutrient status (N, P, K, Ca) of saline soils [43]. The porous structure of biochar allows it to absorb cationic ions, keeping them in the soil and preventing leaching, which is particularly beneficial in salty soils where nutrient leaching is common [44]. These processes increase the concentration of cations in the soil and enhance the electrical conductivity of the soil during the experiment [45]. Furthermore, biochar can raise the CEC of the soil. Since, K is a cation, increased CEC can improve the availability and retention of K ions in the soil [46]. Biochar has also been reported to mitigate the impact of salinity stress on plants by improving soil structure and water retention, thereby facilitating better potassium uptake [47]. However, the effect of biochar on K availability in salty soil can vary depending on factors such as the type of biochar used, the rate of application, soil properties, and the surrounding environment [48]. It is crucial to understand the impacts precisely, because impact can vary depending on the kind of biochar used, rate of application and the properties of the soil [49,50]. To gain a comprehensive understanding of the effect of biochar on K availability in saline soils, site-specific studies are recommended. The application of biochar in saline soil can increase microbiological activity, thereby

mediating plant nutrient availability and uptake kinetics [51]. Increased microbial activity facilitated by biochar application can aid in the liberation of K from minerals and organic debris in the soil, rendering it more accessible to plants. Additionally, biochar can influence soil pH. Some types of biochar exhibit liming properties, which can help alleviate the acidic conditions often found in saline soils [52]. Wakeel [53] indicated that potassium is more soluble in neutral to slightly acidic environments; a more neutral pH can increase K availability. When biochar and FYM are used together, the availability of nutrients in saline soil can be improved. The microbial activity, nutrient retention, and soil structure are all enhanced by these amendments [54]. The FYM has a lot of organic materials have increased the soil's ability to retain nutrients, aids in the development of stable aggregates, and encourages microbial activity [55]. Apart from these, biochar is having higher capacity of pores, prevent the leaching loss of nutrient and maintains the concentration of nutrients in soil solution. Combined application of FYM and biochar improved the plant nutrient concentration, microbial parameters and physical condition of soils [56]. Combined use of NPK along with FYM and pressmud improved soil nutrient status, organic carbon fractions and carbon management index under pigeonpea-wheat cropping system [57] and soil hydro-physical properties [20]. FYM, VC along with PSB also solubilized P from Fe-P and Al-P [58].

## 3.2. Effect on soil microbial parameters

The effect of different organic sources supplied through PM, FYM, and BC were evaluated for the soil enzymatic activities over a period of one year (Table 3). Increasing the organic matter content in treatment improved the soil enzymatic and microbial activity over control and RDF treatments. Alkaline phosphatases (ALPs) were ranged from 86.3 (control) to 147.7 µg PNP/g soil/h ($T_7$-FYM 5 t/ha + BC 5 t/ha). The added organic sources (10 t/ha) by equally comprised of FYM and BC recorded the highest value of ALPs. Other treatments $T_8$ (PM 5 t/ha + BC 5 t/ha) and $T_6$ (FYM 5 t/ha + PM 5 t/ha) also equally improved the ALPs at 5 percent level of significance ($p = 0.05$). Similarly, treatment $T_9$ (129.1 µg PNP/g soil/h) and $T_5$ (133.7 µg PNP/g soil/h) were at par with each other in enhancing ALPs over control. The comparison of data of the highest ALPs in treatment $T_7$ with the RDF treatment showed 60.4 percent improvement; and 71.1% improvement over the control treatment. The arylsulphatase activities was also analysed in all the treatments of the experiment and the highest value was found in the $T_7$ (application of FYM 5 t/ha + BC 5 t/ha) as well as $T_8$ (application of PM 5 t/ha + BC 5 t/ha) treatments. The value of the arylsulphatase activities ranged from 97.7 to 212.7 µg PNP/g soil/h. The arylsulphatase activities in different treatments were 212.7, 211.6, 194.2, 180.2, 169.5,

**Table 3. Effect of different treatments on soil microbial parameters.**

| Treatments | | Alkaline phosphatase activity | Arylsulphatase activity | DHA | MBC |
|---|---|---|---|---|---|
| | | µg PNP/g soil/h | µg PNP /g soil/h | µg TPF/g soil/h | mg/kg |
| $T_1$ | Control | 86.3 | 97.7 | 21.1 | 86.3 |
| $T_2$ | RDF | 92.1 | 139.2 | 28.5 | 92.0 |
| $T_3$ | FYM (10 t/ha) | 115.7 | 158.0 | 34.1 | 115.6 |
| $T_4$ | PM (10 t/ha) | 122.2 | 162.3 | 35.7 | 122.2 |
| $T_5$ | BC (10 t/ha) | 133.7 | 169.5 | 36.6 | 133.5 |
| $T_6$ | FYM (5 t/ha) + PM (5 t/ha) | 136.0 | 180.2 | 47.8 | 136.1 |
| $T_7$ | FYM (5 t/ha) + BC (5 t/ha) | 147.7 | 212.7 | 51.9 | 147.8 |
| $T_8$ | PM (5 t/ha) + BC (5 t/ha) | 139.6 | 211.6 | 43.1 | 139.4 |
| $T_9$ | FYM (5 t/ha) + BC (2.5 t/ha) + PM (2.5 t/ha) | 129.1 | 194.2 | 38.4 | 129.0 |
| LSD ($p = 0.05$) | | 5.9 | 8.8 | 1.8 | 6.0 |

162.3, 158.0, 139.2 and 97.7 µg PNP/g soil/h in $T_7$ (FYM 5 t/ha + BC 5 t/ha), $T_8$ (PM 5 t/ha + BC 5 t/ha), $T_9$ (FYM 5 t/ha + BC 2.5 t/ha + PM 2.5 t/ha), $T_5$ (BC 10 t/ha), $T_4$ (PM 10 t/ha), $T_3$ (FYM 10 t/ha), $T_2$ (RDF) and $T_1$ (control), respectively. The DHA is an indicator of transformation of organic form of plant nutrients to inorganic form. In this experiment, addition of 10 t/ha organic sources through FYM and BC showed the maximum level of DHA (51.9 µg TPF/g soil/h); whereas, other organic treatments also showed significant amount of DHA over the control treatment (21.1 µg TPF/g soil/h) and RDF (28.5 µg TPF/g soil/h). The different combinations of FYM, PM, BC were showed the higher DHA level as compared to the alone application of FYM, PM, and BC. The MBC was measured and found maximum (147.8 mg/kg) in the treatment $T_7$ (FYM 5 t/ha + BC 5 t/ha); subsequently in the treatment $T_8$ comprised with PM + BC, and in $T_6$ (FYM + PM) as 139.4 mg/kg and 136.1 mg/kg, respectively. The MBC was significantly improved by the addition of RDF over the control treatment. However, increasing the organic C through the addition of organic substances like FYM, PM, and BC in saline soil had edge over RDF in improving the microbial activities in the soil.

It was also observed that biochar applied treatments might have supplied the organic C for longer period; and FYM might be the instant source of C for soil microbial population. According to Liao et al. [59], biochar increased the activity of nearly all soil enzymes and DHA by 22% in soil. DHA, a critical intracellular enzyme found among microbes, plays a pivotal role in regulating the transfer of electrons from organic substances to inorganic acceptors. This process significantly impacts the decomposition of organic matter within the soil [23]. Numerous investigations have explored the impact of incorporating biochar on soil enzyme activities, and the conflicting findings such as increased [60], unchanged [61], and decreased [62] microbial enzyme activities in biochar-amended soils have been reported. The relationship between biochar and soil enzyme activity is complex, as biochar can modify soil-related factors, thus directly and indirectly affecting soil enzyme kinetics. The diverse responses of intra- and extracellular microbial enzyme activities to biochar amendment are influenced by factors such as the type of biochar feedstock [63], pyrolysis temperature [64], soil characteristics [60], and the specific types of enzymes involved [65]. Nevertheless, as of now, the precise mechanism remains to be fully elucidated. The MBC acts as a measurable indicator of labile carbon in the soil, providing valuable insights into biological activity. Its levels are primarily influenced by factors such as soil organic carbon, water retention capacity, and soil pH. The introduction of biochar into the soil leads to a significant increase in soil pH, improved water holding capacity, and elevated levels of soil organic carbon [66]. This augmentation leads to a notable rise in both MBC and microbial biomass nitrogen (MBN) in soils subjected to biochar amendment [67]. Hua et al. [68] have investigated the impact of modified biochar on changes in microbial biomass carbon (MBC) in soil. The findings revealed a significant increase of 24.58% in MBC with the application of biochar to the soil. The metabolic activities of microbial biomass are crucial in controlling decomposition and virtually every reaction in the soil N cycle. Additionally, microbial biomass contributes to the formation of soil organic N by stabilizing necromass and microbial byproducts [69]. The microbial biomass N is recognized as the most labile organic nitrogen fraction in soil, soil microbial biomass can both produce and consume $NH_4^+$ and $NO_3^-$ [70]. These findings are clearly correlated with the findings of the current experiment that available N is more in organic substances applied soil particularly in biochar applied treatments.

## 4. Conclusions

Salts ion concentration in soils often face limitations in plant nutrient supply for crop growth. However, sizable areas affected by salinity in India hold significant potential for feeding the

growing population. Excessive salt content can be minimized through the application of organic matter, which helps reduce salt toxicity, improve nutrient availability, and promote better establishment of crop plants. In this incubation experiment, three types of organic substances were compared with a control (without fertilizer), organic manure and the recommended dose of fertilizer (supplied through chemical fertilizer). Particularly, the application of organic substances, especially biochar combined with FYM, improved soil physico-chemical and biological parameters. The combined application of organic sources showed more pronounced effects compared to the application of a single organic source. The advantages of combined application include the distribution of different pools and forms of C and other nutrients, as well as the physical and chemical nature of the organic sources and reaction rate among others. Result indicated that incubation of organic substance (applied 10 t/ha) improved the available plant nutrients like N, P, K, S and soil enzymatic activities. The organic C and available N was recorded the highest in treatment $T_7$ (FYM 5 t/ha + BC 5 t/ha); whereas, available K and S in treatment $T_5$ (BC 10 t/ha). Application of organic soil amendments comprised treatment $T_7$ (FYM 5 t/ha + BC 5 t/ha) improved the alkaline phosphatases, arylsulphatase activity, DHA and MBC over RDF treatments. In overall, application of organic substance 10 t/ha through BC or with FYM had improved the plant nutrient availability and microbial activities in soil. These findings will contribute to understanding the reclamation of degraded soils through the utilization of various organic wastes.

## 4.1. Future line of work

Further research particularly site-specific investigations are needed to explore the use of different locally available and cost-effective organic wastes in addressing soil degradation across various production systems. Conducting such studies will provide valuable insights into the effectiveness of different organic waste materials in soil reclamation and help offering tailored solutions to specific agricultural contexts.

## Acknowledgments

Authors are thankful to the technical officer of the Natural Resource Management Unit, ICAR-DRMR, Bharatpur for technical help during the course of investigation.

## Author Contributions

**Conceptualization:** M. L. Dotaniya, M. D. Meena, Brij Lal Lakaria, R. S. Jat, P. K. Rai, Kuldeep Kumar, Harpreet Singh.

**Data curation:** Brij Lal Lakaria.

**Formal analysis:** M. L. Dotaniya, M. D. Meena, V. D. Meena, Brij Lal Lakaria, R. S. Jat, Kuldeep Kumar, R. K. Doutaniya, Harpreet Singh.

**Funding acquisition:** M. D. Meena, V. D. Meena, R. S. Jat, P. K. Rai.

**Investigation:** M. L. Dotaniya, R. L. Choudhary, V. D. Meena, Brij Lal Lakaria, R. K. Doutaniya.

**Methodology:** M. L. Dotaniya, M. K. Meena, V. D. Meena, Brij Lal Lakaria, P. K. Rai.

**Project administration:** M. D. Meena, R. L. Choudhary, Brij Lal Lakaria, R. S. Jat, P. K. Rai.

**Resources:** M. L. Dotaniya, M. D. Meena, M. K. Meena, Harvir Singh, R. S. Jat, P. K. Rai.

**Software:** R. L. Choudhary, M. K. Meena, Harvir Singh, P. K. Rai.

**Supervision:** M. L. Dotaniya, R. L. Choudhary, V. D. Meena, Harvir Singh, Brij Lal Lakaria, R. S. Jat, P. K. Rai.

**Validation:** M. L. Dotaniya, M. D. Meena, R. L. Choudhary, M. K. Meena, V. D. Meena, Harvir Singh, Brij Lal Lakaria, P. K. Rai, R. K. Doutaniya.

**Visualization:** M. D. Meena, M. K. Meena, V. D. Meena.

**Writing – original draft:** M. L. Dotaniya, Kuldeep Kumar, R. K. Doutaniya, Harpreet Singh.

**Writing – review & editing:** M. L. Dotaniya, Kuldeep Kumar, R. K. Doutaniya, Harpreet Singh.

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
