## [Decision Letter · Decision Letter 0]

9 May 2024

PONE-D-24-12566Mineralization kinetics of major plant nutrients in saline soil influenced by application of biochar and organic wastePLOS ONE

Dear Dr. Dotaniya,

Thank you for submitting your manuscript to PLOS ONE. After careful consideration, we feel that it has merit but does not fully meet PLOS ONE’s publication criteria as it currently stands. Therefore, we invite you to submit a revised version of the manuscript that addresses the points raised during the review process. Please also submit a list with answers to all points raised by the reviewer.

We look forward to receiving your revised manuscript.

Kind regards,

Paulo H. Pagliari

Academic Editor

PLOS ONE

Journal Requirements:

Reviewers' comments:

Reviewer's Responses to Questions

**Comments to the Author**

1. Is the manuscript technically sound, and do the data support the conclusions?

Reviewer #1: No

Reviewer #2: Yes

Reviewer #3: Yes

Reviewer #4: Partly

Reviewer #5: Partly

2. Has the statistical analysis been performed appropriately and rigorously? 

Reviewer #1: I Don't Know

Reviewer #2: Yes

Reviewer #3: Yes

Reviewer #4: Yes

Reviewer #5: Yes

3. Have the authors made all data underlying the findings in their manuscript fully available?

Reviewer #1: Yes

Reviewer #2: Yes

Reviewer #3: Yes

Reviewer #4: Yes

Reviewer #5: Yes

4. Is the manuscript presented in an intelligible fashion and written in standard English?

Reviewer #1: Yes

Reviewer #2: Yes

Reviewer #3: Yes

Reviewer #4: Yes

Reviewer #5: No

5. Review Comments to the Author

Reviewer #1: The authors have conducted good research work however; the MS needs major improvement before it can consider for publication. I have provided below some shortcomings identified in the MS and they need to be addressed.

General comments

1. The research question is not completely clear throughout the introduction, which also lacks a clear hypothesis. Please present a hypothesis. Remember that the research objectives must result from the research hypothesis.

2. Please, could you clarify the use of @ symbol to express the different treatments applied? I suggest removing it and replacing it.

3. A deeper scientific interpretation of your findings in the results and discussion section is strongly suggested. The novelty and implication of your study should also be highlighted throughout this section. Paragraphs should be better structured and interconnected. The discussion should link all of your findings.

4. A systematic description of your results must be done, highlighting for the reader observations that are most relevant to the topic under investigation and it can be used in the discussion later.

5. The conclusions section should illustrate the mechanistic links of findings obtained under applied treatments. The authors should avoid repeating what has already presented in results and discussion. Please, avoid using abbreviations and acronyms in the conclusions section. Remember that the conclusions must be self-explanatory.

Specific comments

1. What is the method of application of biochar and organic waste?

2. What is the moisture content of soils thought out the incubation study?

3. How to prepared pigeon pea biochar. Mention method and conditions.

4. In general, biochar is alkaline in nature but in this study the pH of biochar is 7.6. Why?

5. What is QA/QC…expand it

6. Mention TOC and nutrient composition of pressmud.

7. Figure 1 needs some correction.

8. In page 1, SOM and C… Elaborate first time appeared

9. In the abstract and table 2, it is mentioned that the addition of organic substances (10 t/ha) didn’t affect the soil pH, whereas in the result and discussion section, it is vice versa (the application of different organic and inorganic treatments significantly affected the soil pH).

10. The overall quality of the MS could be enhanced after addressing those issues

Reviewer #2: MS" Mineralization kinetics of major plant nutrients in saline soil influenced by application of biochar and organic waste" falls under the aim and scope of the journal. it is written in scientific way.

it is having minor suggestions

Revised tile as per the study finding.

strengthen the M&M part

Please check reference list.

Reviewer #3: Introduction

Identify research gaps and highlight importance of the present investigation.

Specify clearly objective of the investigation.

Add this para in introduction

Combined use of NPK along with FYM and Pressmud improved soil nutrient status, organic carbon fractions and carbon management index under pigeonpea-wheat cropping system (Mandal et al., 2013) and soil hydro-physical properties (Mandal et al., 2018). FYM, VC along with PSB also solubilized P from Fe-P and Al-P (Bairwa et al., 2021).

Bairwa, R., Chattopadhyay, N., Mandal, N. and Singh, M. (2021) Phosphorus Solubilization under Organic and Inorganic Sources of P in Red and Alluvial Soils and Estimation of Phosphorus Solubilizing Power. Journal of the Indian Society of Soil Science, 69, 440-450

Mandal, N., Dwivedi, B.S., Datta, S.P., Meena, M.C., Tomar, R.K. (2018). Soil hydrophysical properties under different nutrient management practices, their relationship with soil organic carbon fractions and crop yield under pigeonpea-wheat sequence. Journal of Plant Nutrition. 42: 384-400. https://doi.org/10.1080/01904167.2018.1556295 .

Mandal, N., Dwivedi B.S., Meena, M.C., Singh, D., Datta, S.P., Tomar, R.K. and Sharma, B.M. (2013). Effect of farmyard manure, sulphitation pressmud and pigeonpea leaf-litter on soil organic carbon fractions, mineral nitrogen and crop yields in a pigeonpea-wheat cropping system. Field Crops Research 154: 187-178.

Materials and Methods

Section 2.3 : Mention details of incubation temperature and moisture content of soil maintained during laboratory incubation experiment.

Conclusions

Revise conclusion part. Highlight specific recommendation in quantitative terms. Give future line of work in this direction.

Reviewer #4: Revise the manuscript, considering the given suggestions. If possible, then include more data and a sound discussion to justify the requirement of a good quality publication. Critically check the abstract and conclusion; it has some contradictory statements.

Reviewer #5: The authors have used Mineralization kinetics word in the manuscript title, however there is no kinetics involved in result or conclusion. The title misleads that first or second order kinetics has been for mineralization study. Further, the results has no data on mineralization or nutrient dynamics. The results of incubation study presented is only for the change in initial and final concentration of NP S and OC. Its a simple experiment with no new information as its a well known fact that addition of organic matter in any form would certainly enhance nutrient content in soil especially when there is no crop under study ie an incubation experiment.

6. PLOS authors have the option to publish the peer review history of their article (what does this mean?). If published, this will include your full peer review and any attached files.

Reviewer #1: No

Reviewer #2: No

Reviewer #3: No

Reviewer #4: No

Reviewer #5: No

---

## [Author Response · Author response to Decision Letter 0]

10 Jun 2024

Revised MS submitted with a letter of response.

---

## [Decision Letter · Decision Letter 1]

8 Jul 2024

Dynamics of major plant nutrients and soil enzymatic activities in saline soil influenced by application of biochar and organic waste

PONE-D-24-12566R1

Dear Dr. Dotaniya,

We’re pleased to inform you that your manuscript has been judged scientifically suitable for publication and will be formally accepted for publication once it meets all outstanding technical requirements.

Kind regards,

Paulo H. Pagliari

Academic Editor

PLOS ONE

Additional Editor Comments (optional):

Reviewers' comments:

Reviewer's Responses to Questions

**Comments to the Author**

1. If the authors have adequately addressed your comments raised in a previous round of review and you feel that this manuscript is now acceptable for publication, you may indicate that here to bypass the “Comments to the Author” section, enter your conflict of interest statement in the “Confidential to Editor” section, and submit your "Accept" recommendation.

Reviewer #1: All comments have been addressed

Reviewer #2: All comments have been addressed

Reviewer #3: All comments have been addressed

2. Is the manuscript technically sound, and do the data support the conclusions?

Reviewer #1: Yes

Reviewer #2: Yes

Reviewer #3: Yes

3. Has the statistical analysis been performed appropriately and rigorously? 

Reviewer #1: Yes

Reviewer #2: Yes

Reviewer #3: Yes

4. Have the authors made all data underlying the findings in their manuscript fully available?

Reviewer #1: Yes

Reviewer #2: Yes

Reviewer #3: Yes

5. Is the manuscript presented in an intelligible fashion and written in standard English?

Reviewer #1: Yes

Reviewer #2: Yes

Reviewer #3: Yes

6. Review Comments to the Author

Reviewer #1: I thank the authors to largely address my comments. Therefore, I think the MS can be accepted in its present form.

Reviewer #2: Authors incorporated the suggestions. Now it is okay for addressing scientific communities. It recommended for future volume of the journal.

Reviewer #3: Authors have satisfactorily addressed all issues raised by me. The manuscript may be accepted in its present form.

7. PLOS authors have the option to publish the peer review history of their article (what does this mean?). If published, this will include your full peer review and any attached files.

Reviewer #1: No

Reviewer #2: No

Reviewer #3: No

---

## [Editor Report · Acceptance letter]

16 Sep 2024

PONE-D-24-12566R1 

PLOS ONE

Dear Dr. Dotaniya, 

I'm pleased to inform you that your manuscript has been deemed suitable for publication in PLOS ONE. Congratulations! Your manuscript is now being handed over to our production team.

Kind regards, 

on behalf of

Dr. Paulo H. Pagliari 

Academic Editor

PLOS ONE